# Muslim undergraduate biology students' evolution acceptance in the United States

**M. Elizabeth Barnes** [1]*, **Julie A. Roberts**[2], **Samantha A. Maas**[2], **Sara E. Brownell**[2]

**1** The Social Perceptions of Science Lab, Department of Biology, Middle Tennessee State University, Murfreesboro, Tennessee, United States of America, **2** The Biology Education Research Lab, Research for Inclusive STEM Education Center, School of Life Sciences, Arizona State University, Tempe, Arizona, United States of America

* liz.barnes@mtsu.edu

**Data Availability Statement:** All relevant data are within the paper and its S1 File.

**Funding:** MEB and SEB, National Science Foundation (nsv.gov) IUSE #1818659. The funders had no role in study design, data collection and

## Abstract

Evolution is a prominent component of biology education and remains controversial among college biology students in the United States who are mostly Christian, but science education researchers have not explored the attitudes of Muslim biology students in the United States. To explore perceptions of evolution among Muslim students in the United States, we surveyed 7,909 college students in 52 biology classes in 13 states about their acceptance of evolution, interest in evolution, and understanding of evolution. Muslim students in our sample, on average, did not agree with items that measured acceptance of macroevolution and human evolution. Further, on average, Muslim students agreed, but did not strongly agree with items measuring microevolution acceptance. Controlling for gender, major, race/ethnicity, and international status, we found that the evolution acceptance and interest levels of Muslim students were slightly higher than Protestant students and students who are members of the Church of Jesus Christ of Latter-day Saints. However, Muslim student evolution acceptance levels were significantly lower than Catholic, Jewish, Buddhist, and Hindu students as well as students who did not identify with a religion (agnostic and atheists). Muslim student understanding of evolution was similar to students from other affiliations, but was lower than agnostic and atheist students. We also examined which variables are associated with Muslim student acceptance of evolution and found that higher understanding of evolution and lower religiosity are positive predictors of evolution acceptance among Muslim students, which is similar to the broader population of biology students. These data are the first to document that Muslim students have lower acceptance of evolution compared to students from other affiliations in undergraduate biology classrooms in the United States.

## Introduction

Evolution is a foundation of biology that should be taught at every level of biology education [1–3], yet it remains a controversial scientific theory among the public [4] and college biology students [5–8]. Religious identity and beliefs are a major source of rejecting evolution [9–11], but the evolution acceptance literature in the context of the United States is dominated by the

analysis, decision to publish, or preparation of the manuscript.

**Competing interests:** The authors have declared that no competing interests exist.

study of Christian students because the vast majority of religious students in biology classes are Christian [5]. However, we currently know very little about evolution acceptance, evolution understanding, and interest in evolution among biology students from other religious affiliations, including Muslim students, in the United States. Further, we do not know if the same variables associated with evolution acceptance among the broader population of students (for example, religiosity and understanding of evolution) are the same for students of other religious affiliations as they are for Christians. In this current study, we extend beyond focusing on Christian biology student evolution perceptions in the United States and examine evolution acceptance, evolution understanding, and interest in evolution among Hindu, Buddhist, and Muslim college biology students across the United States, as well as the variables associated with their evolution acceptance levels. Based on prior data collected outside of the United States, we would predict that Muslim students in the United States may be less receptive to evolution, so we focus the manuscript on this population of students.

## Studies of acceptance of evolution around the globe suggest Muslim student acceptance of evolution is low

In Muslim majority countries, studies have demonstrated low rates of evolution acceptance among students and the public. In Turkey, researchers have found that college students training to be biology teachers scored low on the Measure of Acceptance of the Theory of Evolution (MATE) [12, 13]. Among Lebanese, Egyptian Sunni, and Shiite Muslims in high school, 23% of students agreed that "evolution is scientifically wrong" and 26% were undecided about the scientific validity of evolution [14]. Among Muslim medical students in Pakistan, 68% rejected evolution based on their religious beliefs, and these students also showed a low understanding of evolution [15]. Among 18 Pakistani high school teachers who were interviewed about their views on evolution, researchers reported that almost all of the teachers rejected human evolution due to a perceived conflict with their religious beliefs [16]. The conclusion that Muslim evolution acceptance is low has also been found in studies with large sample sizes across many nations. In 2015, researchers surveyed over 10,000 teachers across 30 countries about their perceptions of evolution and found that among all majority Muslim countries they surveyed (Algeria, Morocco, Senegal, Lebanon, and Tunisia), 70% or more of their teachers subscribed to special creationism as opposed to evolution [17]. In Indonesia, where the largest Muslim population in the world resides, Rachmatullah and colleagues found that the MATE scores of pre-service biology teachers were lower than those of pre-service teachers in non-Muslim dominant countries [18]. These studies collectively demonstrate that outside of the United States, levels of evolution acceptance for Muslim individuals are low and that a perceived conflict between religion and evolution is a factor.

Muslim student acceptance of evolution has been shown to be low compared to the evolution acceptance of those with other religious identities. When comparing Muslim and Christian biology students' views on evolution at a Lebanese university, Muslim students were much less likely than Christian students to support evolution [19]. When comparing Greek and Turkish students' evolution acceptance scores on the MATE, researchers found that Greek students who were mostly Christian scored approximately two standard deviations higher on acceptance of evolution than Turkish students who were mostly Muslim [20]. Comparisons of Muslim and Christian secondary teachers in Malaysia and South Africa revealed that Muslim teachers were much more likely to subscribe to special creationism than Christian teachers [21, 22]. In England, almost all Muslim 14–16-year-old students believed humans were created in their current form whereas only half of Christian students reported the same belief [23] and Muslim residents were least accepting of the evolution of plants, animals, and

humans compared to Christians and non-religious individuals [24]. In the United States, the evolution acceptance of Christian students is low compared to non-religious students [4, 25], but no one has explored perceptions of evolution among Muslim students in the United States. This gap in the literature may make it challenging for evolution instructors to consider the needs of Muslim students at U.S. colleges and universities.

## What is known about evolution acceptance among Muslim individuals in the United States?

Muslim students in the United States may have similarly low levels of evolution acceptance as individuals from Muslim majority countries because they are affiliated with the same religion. In Islam, the Quran depicts a special creationist origin of humans similar to that of the Christian Bible [26]. A literal interpretation of the Bible in Christianity has been identified as one of the major sources of rejection of evolution in the United States [27] so we might expect that Muslim individuals who hold a strict literal interpretation of their religious text, regardless of their country of residence, will also have low acceptance of evolution. However, there are some reasons that Muslim individuals might have higher evolution acceptance in the United States.

Sociological public polls show that American Muslims are less likely than Muslims in other countries to believe that the Quran should be read literally and more likely to believe that the Quran can be interpreted multiple ways [28, 29]. Thus, American Muslim students may be more likely to interpret depictions of human creation in the Quran as a symbolic story and thus be able to accommodate an acceptance of evolution. Further, Muslim individuals in the United States have an average of eight more years of formal education compared to Muslims globally [30]. Since higher education levels are associated with higher evolution acceptance [25], we may expect that Muslim students in the United States will be more accepting of evolution, particularly among college biology students who have likely had more formal education than many Muslim individuals in other countries. One study in Canada, which is more reflective of the United States population than majority Muslim countries, found that Muslim high school science teachers largely accepted evolution of organisms except for humans [16]. Among students in the United States, acceptance of human evolution has been shown to be a separate psychological construct from acceptance of microevolution and human evolution [31], so it may be the case that acceptance of evolution among Muslims in the United States depends on the context of evolution and whether it includes humans.

## Is understanding of evolution and religiosity related to evolution acceptance levels of Muslim students?

The extent to which religiosity and understanding of evolution are related to acceptance of evolution has been shown to be variable across studies [10, 11, 25, 32–36], across scales of evolution (microevolution, macroevolution, human evolution, and common ancestry of life) [25], and may be different across different populations of students [37]. Knowing what variables influence student acceptance of evolution has implications for the extent to which instructors may want to account for student religious beliefs when teaching evolution. For instance, if religiosity is related to Muslim student evolution acceptance to a greater extent than understanding of evolution, then that may indicate that in addition to traditional instruction aimed at increasing understanding of evolution, these students would benefit from opportunities to learn about the relationship between religion and evolution and where religion and evolution can be compatible versus where they are in conflict. The evolution education literature thus far has not explored whether the common variables associated with evolution acceptance of the broader population of American students, who are predominantly Christian, are also

associated with evolution acceptance among Muslim students. If we do not explore Muslim students independently from other students, we might make assumptions about factors related to their evolution acceptance that are not true.

### The current study and research questions

In this study, we aimed to explore levels of evolution acceptance, interest in evolution, and understanding of evolution among Muslim biology students in the United States. We document the comparisons between these variables among Muslim students and students from other religious affiliations, including Christians, Jewish individuals, Buddhists, Hindus, and individuals who are not religious. Finally, we explore the extent to which variables associated with evolution acceptance in the broader American population are also associated with acceptance of evolution among these students, including Muslims. Our specific research questions are:

1. Controlling for major, gender, race/ethnicity, and international status, do the average levels of evolution understanding, interest in evolution, and evolution acceptance of Muslim students in the United States differ from students who are Protestant, Catholic, Latter-day Saints (LDS), Jewish, Hindu, Buddhist, agnostic, or atheist?

2. To what extent does understanding of evolution and religiosity contribute to students' evolution acceptance levels for each religious affiliation?

## Methods and analyses

We surveyed students in 52 college biology classes at 22 institutions across 13 U.S. states (Alabama, Arizona, California, Florida, Hawaii, Minnesota, North Carolina, New York, Oklahoma, South Carolina, Texas, Utah, Wisconsin) in fall 2018, spring 2020, and fall 2020. Students were recruited through their instructors, who agreed to forward the survey to students before any evolution instruction occurred in the class and offer a small amount of extra credit to students who took the survey. This study was approved by Arizona State University's Institutional Review Board, protocol #8191. Students indicated their consent to participate by clicking a box on the online survey.

### Survey measures

The survey was conducted as part of a larger study on the impact of evolution education on undergraduate biology students. For this study, we included the variables of religious denomination, religiosity, acceptance of evolution, interest in evolution, understanding of evolution, major, gender, race/ethnicity, and international status. All questions used in the analyses can be found in the **S1 File**.

**Religious denomination.** Students were asked to choose a religious affiliation with which they most closely identified and were then grouped into the following categories based on their responses: Muslim, Christian–Catholic, Christian–Protestant/nondenominational, Christian–The Church of Jesus Christ of Latter-day Saints (LDS), Jewish, Hindu, Buddhist, agnostic, and atheist. Students who chose a religion that was not part of a large enough group for statistical analyses were not included in the results of this manuscript (for example, Sikh, Pagan, Satanist, spiritualist, Taoist, Christian–orthodox, etc.). Non-denominational Christians were grouped with Protestant Christians because these groups were closely aligned in our analyses. In all analyses, Muslim students are the reference group because they are the focus of this study.

**Religiosity.**   We measured student religiosity using a scale previously validated with college students [38]. The measure consisted of four items with Likert response options that measure two important components of religiosity: the intrinsic strength of one's religious identity and participation in religious activities. This measure is similar to other common measures used in both studies of religion [39, 40] and studies of evolution acceptance [10, 41]. Further, it was designed to be valid for students across many different religious denominations [38]. Items were aggregated and then divided by four to represent students' average agreement on a Likert scale from 1 (strongly disagree)– 5 (strongly agree) ($\alpha$ = .90).

**Acceptance of evolution.**   Acceptance of evolution refers to the extent to which students personally think evolution is valid and can include acceptance of microevolution, acceptance of macroevolution, acceptance of human evolution, and acceptance of the common ancestry of life on Earth [5, 31]. To measure acceptance of microevolution, macroevolution, and human evolution, we used The Inventory of Student Evolution Acceptance (I-SEA) [31], which has been validated with college biology students [42]. Each scale consists of eight items and items from each scale were aggregated and then divided by eight to represent students' average agreement on a Likert scale from 1 (strongly disagree)– 5 (strongly agree) ($\alpha$ (micro) = .84; $\alpha$ (macro) = .85; $\alpha$ (human) = .91).

To determine whether students accepted the common ancestry of life, we used a previously published survey that asks students to choose from nine different views on the relationship between religion and evolution, some of which reflect an acceptance of the common ancestry of life and some of which indicate a belief that a God/god(s) created species separately from one another [5, 43]. Students were categorized as either accepting or not accepting the common ancestry of life and those who chose options that reflected a special creationist view were categorized as not accepting the common ancestry of life.

**Interest in evolution.**   We developed four items to measure students' interest in evolution because interest can be a strong indicator of motivation to learn a topic [44, 45] and no prior survey existed to measure this variable when we did this study. We measured students' interest in (1) taking a course on evolution, (2) doing undergraduate research on evolution, (3) studying evolution as part of their career, and (4) becoming an evolutionary biologist. Students answered each question on a scale from 0 (not at all) to 10 (very much). We conducted cognitive interviews [46, 47] with 25 undergraduate biology students and revised the questions so that they were being interpreted correctly and so that the wording was not confusing for students. This measure was only included in fall and spring 2020 collections. ($\alpha$ = .88).

**Understanding of evolution.**   Understanding of evolution is different from acceptance of evolution and refers to the extent a student has a good conceptual grasp of current evolutionary theory. A student can have a good understanding of evolution, and yet still choose to not accept evolution [48]. To measure students' evolution understanding, we used two subscales on the previously published Evolutionary Attitudes and Literacy instrument (EALS) [49]. We only used the two subscales (13 items) from the instrument that measure "Evolutionary Knowledge" (e.g., "In most populations, more offspring are born than can survive") and "Evolutionary Misconceptions" (e.g., "Evolution is a linear progression from primitive to advanced species") because these were the subscales related to understanding of evolution. Students were asked to decide whether each item was true or false based on their evolution understanding and were also given an option to indicate "I don't know enough to answer" to avoid correct answers by guessing. We calculated student scores by determining the proportion of correct answers. The EALS has been used in other evolution education studies [10, 50], has shown evidence of reliability and validity among college students [49], and importantly, the items do not appear to conflate evolution acceptance with

evolution understanding [25] ($\alpha$ = .58, which is acceptable for a test that measures content knowledge of a domain (see for example, [51], pg. 135–138)).

**Demographics.** To control for potential confounding variables in our analyses we included race/ethnicity, gender, major, and international status in our survey and our analyses. Students were asked to identify their gender as woman, man, or non-binary. In all analyses, man is the reference group. Students were asked to identify their race/ethnicity and were categorized as Asian, Black, Latinx, White, other race, or multiracial. White is the reference group in all analyses. Students were also asked to identify if they were a biology major or a major other than biology. Finally, to determine international status, we asked students if they were born in the United States.

**Analyses.** Only complete student responses were included in the analyses. Less than 5% of data were missing. All analyses were done in SPSS version 26. All data and syntax for analyses are included in the **S1 File**.

We provide tables of the means and standard deviations for outcome variables to illustrate the central tendencies of the raw data. We provide violin plots to illustrate the variability and distribution of the raw data.

To determine if levels of evolution understanding, evolution acceptance, and evolution interest differ between Muslim students and students from other religious denominations, we used multiple linear regressions and controlled for the potentially confounding variables of race/ethnicity, major, gender, and international status in our analyses. We report the standardized coefficients and p-values for comparisons made between Muslim students and students from other religious denominations. In the case of acceptance of common ancestry, which is a binary rather than continuous outcome variable, we used binary logistic regression and report the odds ratios (OR) and p-values for comparisons between Muslim students and students from other religious affiliations. To determine the extent to which evolution understanding and religiosity predict evolution acceptance among students, we selected only students of a particular religious affiliation for analysis and then ran three multiple linear regressions (human, macro, or micro) and one binary logistic regression (acceptance of common ancestry) with the various evolution acceptance measures as the dependent variables and religiosity and understanding of evolution as the predictor variables. To account for potentially confounding variables, we also controlled for gender, major, and international status. There was not enough variation in race/ethnicity to control for this variable in regressions with these analyses. Full regression tables with omnibus statistics, coefficients for all variables, and standard errors for all coefficients in all analyses can be found in the **S1 File**.

## Results

Undergraduate biology instructors sent the survey to approximately 13,100 potential participants and a total of 7,909 college biology students completed the survey (response rate = ~ 60.4%). Of these students, 16.7% identified as Asian, 5.8% as Black, 16.1% as Latinx, 0.4% as Native Islander, 0.5% as Native American, 49.9% as White, 0.1% as another race/ethnicity, and 10.5% as multiracial. Women were 67.2% of the sample, 32.2% were men, and 0.6% were non-binary, which is similar to the broader population of undergraduate biology students [52]. Biology majors were 53.6% of the sample. Muslim students comprised 2.8% of the sample; Muslim populations in the United States is approximately 1.1% of the population so our sample is similar in percentage [53]. For a breakdown of the religious affiliations of students see Table 1 and for the demographics of Muslim students specifically, see Table 2.

**Table 1. The religious affiliations of undergraduate biology students in this study.**

| Religious Affiliation | Study Participants n = 7,909% (n) |
|---|---|
| Muslim | 2.8 (219) |
| Christian–Protestant[a] | 24.7 (1952) |
| Christian–LDS[b] | 9.9 (780) |
| Christian—Catholic | 23.7 (1877) |
| Jewish | 1.9 (153) |
| Hindu | 2.1 (165) |
| Buddhist | 2.2 (173) |
| Agnostic | 25.1 (1984) |
| Atheist | 7.7 (606) |

[a]Includes Protestant and nondenominational Christians.

[b]This group represents those affiliated with the Church of Jesus Christian of Latter-day Saints who prefer to be named as such as opposed to the term "Mormon." We acknowledge this preference and use LDS as an acronym to shorten the name to fit in tables and figures.

## Finding 1: Muslim student understanding of evolution is similar to students from other religious affiliations, but lower than atheist and agnostic students

Muslim student understanding of evolution was lower than agnostic ($\beta$ = .089, p = .003) and atheist students ($\beta$ = .108 p < .001). There were no significant differences between Muslim student understanding of evolution and that of students from any other religious affiliations (p > .11). See Table 3 for raw means and standard deviations of understanding levels of evolution broken down by religious affiliation. See Fig 1 for distribution of understanding of evolution levels broken down by religious affiliation.

**Table 2. The demographics of college biology Muslim student participants in this study.**

| Student Demographic | Muslim Study Participants n = 219% (n) |
|---|---|
| Major | |
| Biology | 64.8 (142) |
| Other Major | 35.2 (77) |
| Gender | |
| Female | 58.0 (127) |
| Male | 42.0 (92) |
| Non-binary | 0.0 (0) |
| Race/ethnicity | |
| Asian | 57.5 (126) |
| Black | 14.2 (31) |
| Latinx | 0.9 (2) |
| Native Islander | 0.9 (2) |
| Multiracial | 8.7 (19) |
| White | 17.8 (39) |
| Place of Birth | |
| United States | 69.4 (152) |
| Other | 30.6 (67) |

**Table 3. The mean and standard deviation of evolution understanding scores disaggregated by religious affiliation.**

| Affiliation | Mean | Standard Deviation |
|---|---|---|
| Muslim | .67 | .16 |
| Christian–Protestant | .69 | .16 |
| Christian–LDS | .71 | .16 |
| Christian—Catholic | .67 | .16 |
| Jewish | .71 | .14 |
| Hindu | .68 | .17 |
| Buddhist | .69 | .17 |
| Agnostic | .72 | .16 |
| Atheist | .75 | .17 |

## Finding 2: Muslim student interest in evolution is higher than Protestant students and students from the Church of Jesus Christ of Latter-day Saints, but lower than that of students from other religious affiliations

Muslim student interest in evolution was higher than Protestant students (β = -.143, p < .001) and students who are members of the Church of Jesus Christ of Latter-day Saints (β = -.079, p < .001), but lower than Jewish students (β = .043, p = .003), Buddhist students (β = .078, p < .001), agnostic students (β = .153, p < .001) and atheist students (β = .123, p < .001). See Table 4 for mean and standard deviation of interest scores disaggregated by affiliation. See Fig 2 for distribution of interest in evolution levels broken down by religious affiliation.

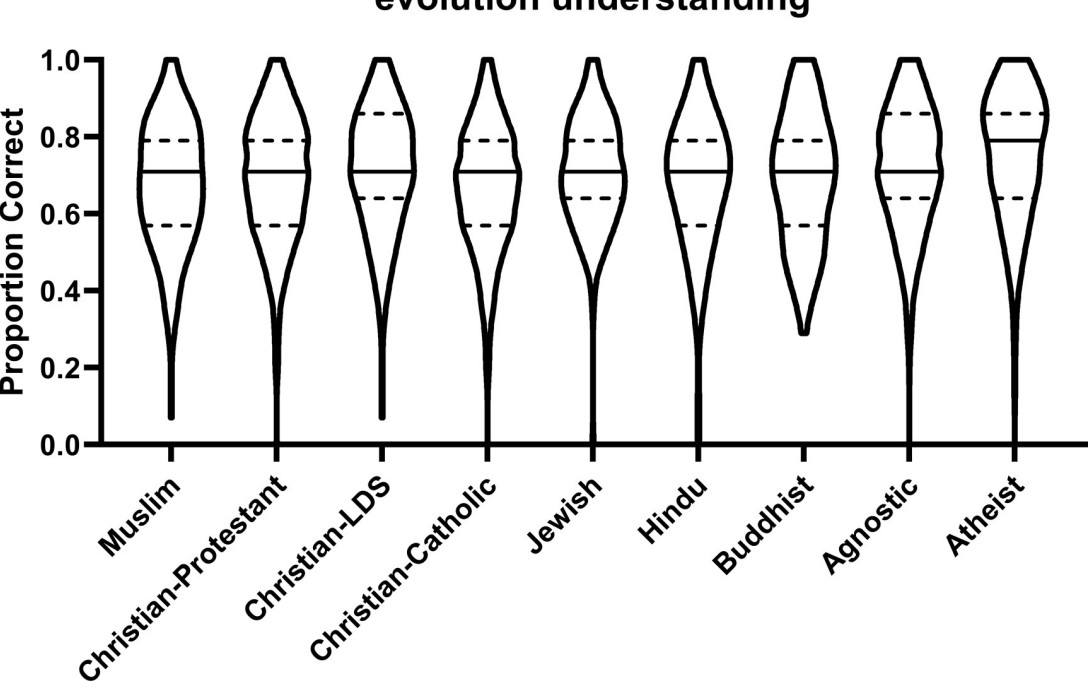

**Fig 1. Evolution understanding.** Violin plots of proportion of correct answers on a test of evolution understanding disaggregated by religious affiliation. The violin shapes are the densities of the data at each point on the y-axis. The solid black lines are the medians, and the top and bottom broken lines are the 75th and 25th percentiles of the data.

**Table 4. The mean and standard deviation of students' aggregated interest in taking an elective course on evolution, interest in conducting research on evolution as an undergraduate, interest in a career involving research on evolution, and interest in becoming an evolutionary biologist.**

| Affiliation | Mean | Standard Deviation |
|---|---|---|
| Muslim | 3.73 | 2.43 |
| Christian—Protestant | 2.69 | 2.26 |
| Christian—LDS | 2.58 | 2.28 |
| Christian—Catholic | 3.83 | 2.31 |
| Jewish | 4.07 | 2.25 |
| Hindu | 4.19 | 2.28 |
| Buddhist | 4.93 | 2.19 |
| Agnostic | 4.43 | 2.34 |
| Atheist | 4.74 | 2.56 |

Scores disaggregated by religious affiliation.

## Finding 3: Muslim student evolution acceptance is higher than Protestant students and students from the Church of Jesus Christ of Latter-day Saints, but lower than that of students from other religious affiliations

**The common ancestry of life on Earth.** Thirty-seven percent of Muslim students chose an option that indicated they accepted the common ancestry of life on Earth. Muslim students were significantly less likely to accept the common ancestry of life on Earth compared to

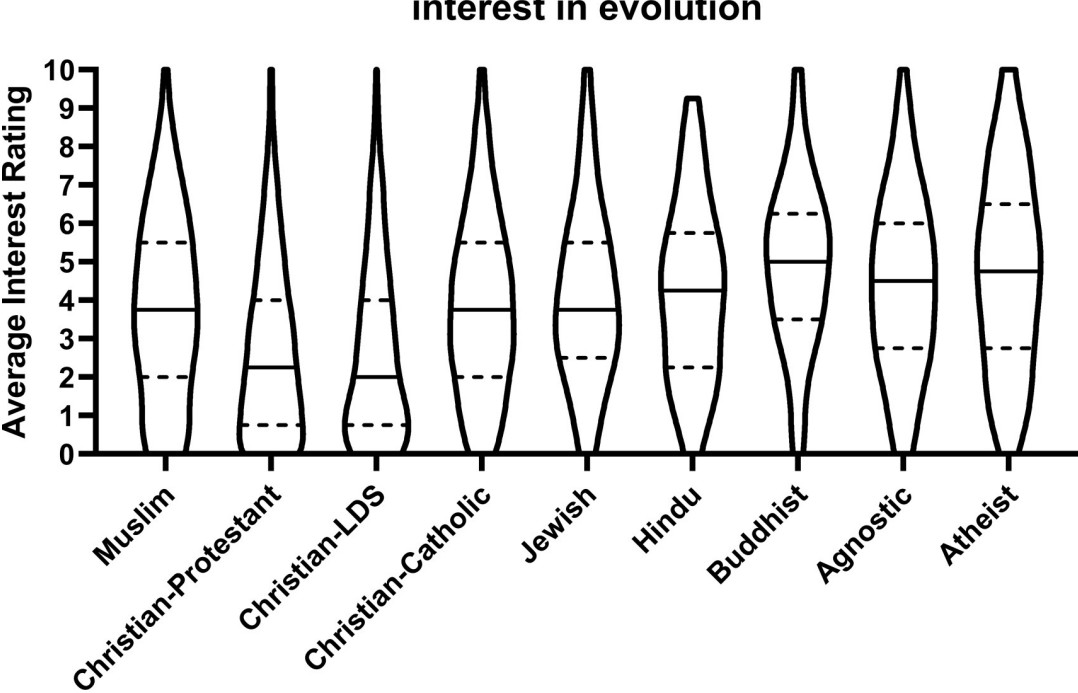

**Fig 2. Interest in evolution.** Violin plots of students' average interest in taking an elective course on evolution, doing undergraduate research on evolution, studying evolution as part of their career, and becoming an evolutionary biologist. The violin shapes are the densities of the data at each point on the y-axis. The solid black lines are the medians, and the top and bottom broken lines are the 75th and 25th percentiles of the data.

Catholic students (OR = 1.96, p < .001), Jewish students (OR = 6.05, p < .001), Hindu students (OR = 6.26, p < .001), Buddhist students (OR = 14.56, p < .001), agnostic students (OR = 32.62, p < .001) and atheist students (OR = 514.63, p < .001). Muslim students were slightly more likely to accept the common ancestry of life compared to students who were members of the Church of Jesus Christ of Latter-day Saints (OR = .68, p = .026). There was no statistically significant difference between Muslim student and Protestant student acceptance of the common ancestry of life (p = .93). Proportions of students that accepted the common ancestry of life disaggregated by religious affiliation can be found in Table 5.

**Human evolution.** Muslim students were less accepting of human evolution compared to Catholic (β = .190, p < .001), Jewish (β = .110, p < .001), Hindu (β = .107, p < .001), Buddhist (β = .126, p < .001), agnostic (β = .371, p < .001), and atheist (β = .301, p < .001) students. Muslim students were more accepting of human evolution than Protestant students (β = -.079, p = .004) and students who were members of the Church of Jesus Christ of Latter-day Saints (β = -.094, p < .001).

**Macroevolution.** Muslim students were less accepting of macroevolution than Catholic students (β = .123, p < .001), Jewish students (β = .075, p < .001), Hindu students (β = .081, p < .001), Buddhist students (β = .092, p < .001), agnostic students (β = .289, p < .001), and atheist students (β = .268, p < .001). Muslim students were more accepting of macroevolution than Protestant students (β = -.098, p = .001) and students who were members of the Church of Jesus Christ of Latter-day Saints (β = -.071, p = .001).

**Microevolution.** Muslim students were less accepting of microevolution than Catholic students (β = .094, p = .002), Jewish students (β = .043, p = .002), Hindu students (β = .051, p < .001), Buddhist students (β = .075, p < .001), agnostic students (β = .242, p < .001), and atheist students (β = .205, p < .001). Muslim student acceptance of microevolution was not statistically different from Protestant students (p = .889) or students who were members of the Church of Jesus Christ of Latter-day Saints (p = .166).

Means and standard deviations of human evolution acceptance, macroevolution acceptance, and microevolution acceptance disaggregated by religious affiliation can be found in Table 6. Distributions of responses, medians, and 1st and 3rd quartile of responses can be found in Fig 3.

Overall, these results indicate that Muslim students are less accepting of evolution compared to Catholic, Jewish, Hindu, Buddhist, agnostic, and atheist students, but are slightly more accepting of evolution compared to Protestant students and students who are members of the Church of Jesus Christ of Latter-day Saints.

**Table 5. Proportion of students that believe life on Earth shares a common ancestor disaggregated by religious affiliation.**

| Affiliation | Accepts common ancestry of life |
|---|---|
| Muslim | 36.5% |
| Christian—Protestant | 33.3% |
| Christian—LDS | 29.9% |
| Christian—Catholic | 48.5% |
| Jewish | 77.8% |
| Hindu | 78.2% |
| Buddhist | 89.0% |
| Agnostic | 94.7% |
| Atheist | 99.7% |

**Table 6. The mean and standard deviation of human evolution acceptance, macroevolution acceptance, and microevolution acceptance scores disaggregated by religious affiliation.**

| | Human Evolution Acceptance | | Macroevolution Acceptance | | Microevolution Acceptance | |
|---|---|---|---|---|---|---|
| **Affiliation** | **Mean** | **SD** | **Mean** | **SD** | **Mean** | **SD** |
| Muslim | 3.35 | .81 | 3.60 | .62 | 4.07 | .60 |
| Christian—Protestant | 3.24 | .89 | 3.48 | .74 | 4.14 | .60 |
| Christian—LDS | 3.16 | .87 | 3.47 | .69 | 4.20 | .54 |
| Christian—Catholic | 3.74 | .65 | 3.82 | .53 | 4.24 | .52 |
| Jewish | 4.07 | .58 | 3.99 | .50 | 4.34 | .52 |
| Hindu | 3.94 | .56 | 3.97 | .51 | 4.25 | .55 |
| Buddhist | 4.04 | .55 | 4.01 | .48 | 4.35 | .55 |
| Agnostic | 4.09 | .58 | 4.07 | .53 | 4.44 | .48 |
| Atheist | 4.33 | .58 | 4.30 | .52 | 4.55 | .48 |

Next, we report results on the evolution acceptance of students and ask which variables are related to their evolution acceptance levels.

## Finding 4: Among Muslim students, a higher understanding of evolution and lower religiosity are positive predictors of evolution acceptance

**The common ancestry of life on Earth.** The variables explained approximately 10% of the variation in whether Muslim students accepted the common ancestry of life (chi-square = 21.847, df = 5, p < .001). A higher understanding of evolution was not a significant predictor of whether a Muslim student would accept common ancestry (p = .310), but higher religiosity was a negative predictor of whether a student would accept common ancestry (OR = .41, p < .001).

**Human evolution.** The variables explained approximately 17% of the variation in human evolution acceptance scores among Muslim students (F(5, 213) = 10.13, p < .001). A higher understanding of evolution was a significant positive predictor of human evolution acceptance scores ($\beta$ = .200, p = .002), but higher religiosity was a stronger negative predictor of human evolution acceptance scores ($\beta$ = -.423, p < .001).

**Macroevolution.** The variables explained approximately 9% of the variation in macroevolution acceptance scores for Muslim students (F(5, 213) = 5.35, p < .001). A higher understanding of evolution was a significant positive predictor of macroevolution acceptance scores ($\beta$ = .267, p < .001) and higher religiosity was a negative predictor of macroevolution acceptance scores ($\beta$ = -.223, p = .001).

**Microevolution.** The variables explained approximately 10% of the variation in microevolution scores for Muslim students (F(5, 213) = 5.80, p < .001). A higher understanding of evolution was a significant positive predictor of microevolution acceptance scores ($\beta$ = .300, p < .001) and higher religiosity was a weaker negative predictor of microevolution acceptance scores ($\beta$ = -.202, p = .003).

Since we had an adequate sample size for students with other religious affiliations to do these analyses, we also ran the same regressions for Hindu, Buddhist, Jewish, Protestant, LDS, and Catholic students (**S1 File**). We found that evolution understanding was related to acceptance of common ancestry, human evolution, macroevolution, and microevolution for almost all religions (with the exception of Jewish student acceptance of common ancestry). We found that religiosity was not related to acceptance of common ancestry among Buddhist students, was not related to acceptance of human evolution, macroevolution or microevolution among

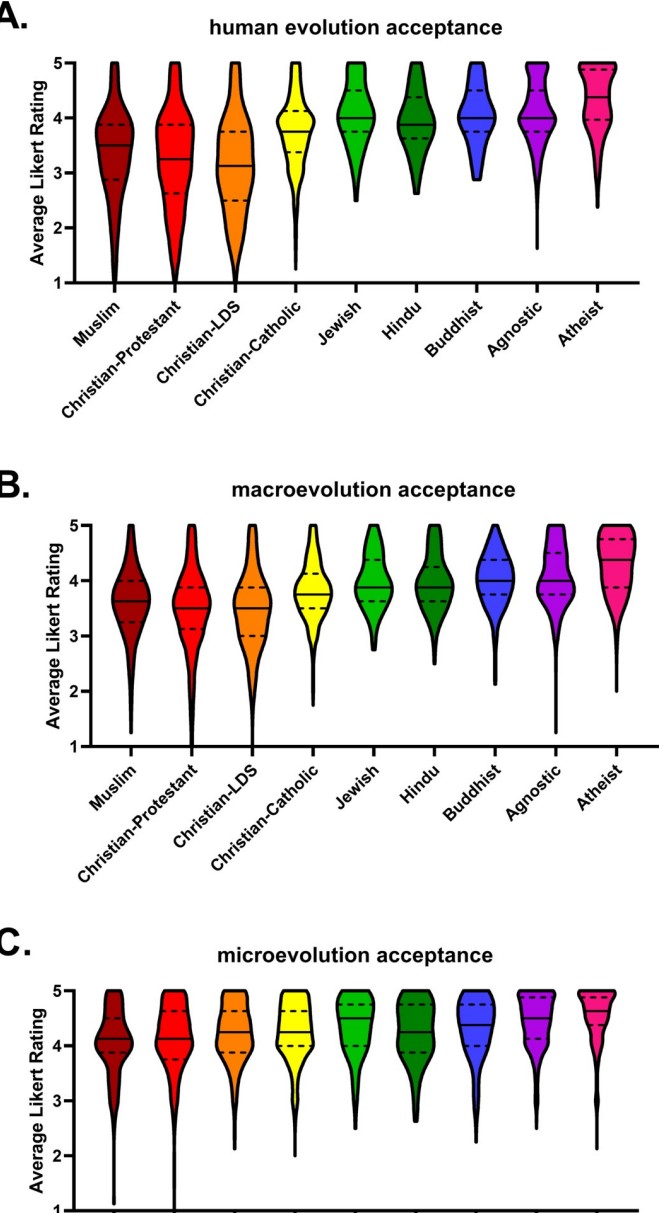

**Fig 3. Acceptance of evolution.** Violin plots of students' (A) human evolution acceptance scores, (B) macroevolution acceptance scores and (C) microevolution acceptance scores disaggregated by religious affiliation. The violin shapes are the densities of the data at each point on the y-axis. The solid black lines are the medians, and the top and bottom broken lines are the 75th and 25th percentiles of the data. When there is not line for the median or 25th percentile this means those values overlap with the minimum value of the scale.

Jewish, Hindu, and Buddhist students, and was not related to microevolution acceptance among Catholic students. However, since the focus of this manuscript is on Muslim students, we only report full results for Muslim students in the main body of the manuscript. Full

regression tables from the analyses of the broader population of students and for students from each religious affiliation can be found in the **S1 File**.

Taken together these results indicate that among Muslim students, controlling for major, gender, and international status, a higher understanding of evolution and lower religiosity are significant positive predictors of evolution acceptance. Further, across different measures of evolution acceptance, higher religiosity is a stronger negative predictor of human evolution acceptance and acceptance of the common ancestry of life on Earth than acceptance of microevolution.

## Discussion

In the first study that we know of that has examined Muslim students in undergraduate biology classes in the United States, we found that Muslim students' evolution acceptance and interest levels are lower than Catholic, Jewish, Hindu, Buddhist, agnostic, and atheist students and were only slightly higher than Protestant students and students who are members of the Church of Jesus Christ of Latter-day Saints. Muslim student acceptance of evolution was particularly low when considering their acceptance of human evolution and the common ancestry of life on Earth; Muslim students, on average, did not agree with items indicating acceptance of human evolution and only 36.5% of Muslim students chose items that indicated acceptance of the common ancestry of life. While past research on student acceptance of evolution in the United States has been focused on Christian students, largely because Christian students are the most prevalent in undergraduate biology classes in the United States, this study implies a need to consider how we can improve evolution acceptance for Muslim students in the United States since their acceptance levels are similarly as low as Christians.

Despite low levels of acceptance of evolution among Muslims students in this study, we did find that compared to Christian students, Muslim student evolution acceptance was slightly higher than Protestant students and lower than Catholic students. In prior studies in which Muslim and Christian evolution acceptance was examined outside of the United States, Christian students were often characterized as much more accepting of evolution than Muslim students. However, the Christians in prior studies were majority Catholic or Eastern Orthodox, both of which have official stances in favor of evolution [19, 20]. In the United States, however, there is a larger population of Protestant Christians, who tend to accept evolution less than those from other denominations [54]. Thus, Muslim students' evolution acceptance may be comparatively higher in the United States because Christians in the United States are more often affiliated with Protestant denominations of Christianity.

We also looked at variables associated with Muslim student evolution acceptance and found that similar to other biology students, a higher understanding of evolution and lower religiosity is associated with more acceptance of evolution. Understanding of evolution was most strongly related to microevolution acceptance compared to human evolution acceptance and acceptance of the common ancestry of life on Earth. Acceptance of microevolution was relatively high among all students, including Muslim students. However, acceptance of common ancestry of life and human evolution acceptance was low among Muslim students and religiosity was a stronger predictor of human evolution acceptance and common ancestry than understanding of evolution.

Since we had data for students from other religious affiliations, we also looked at relationships between understanding of evolution and religiosity among those groups. While we found that evolution understanding was related to acceptance of evolution consistently across

religions, we found that how religious a student is (their religiosity) was not consistently related to evolution acceptance for students from all religious affiliations. Specifically, the strength of students' religiosity was not related to how much they accepted evolution among Jewish, Hindu, and Buddhist students. This is in line with research that shows perceived conflict between religion and evolution varies across religions, which affects the relationship between religiosity and acceptance of evolution [55].

## Acceptance of microevolution and implications for teaching

Microevolution acceptance was consistently high across students from different religions, including Muslim students, and it may be the case that instructors can emphasize the widespread acceptance of microevolution as a gateway to help students accept macroevolution and human evolution. Although novice students may see microevolution, macroevolution, and human evolution as separate phenomena, biologists often see the patterns of macroevolution as a result of accumulation of microevolutionary changes between two populations experiencing reproductive isolation from one another [56]. When two populations of organisms become isolated from one another, either through geographic barriers (allopatric) or behavioral barriers (sympatric), microevolutionary changes accumulate differentially between these two populations, eventually leading to speciation of the two populations (macroevolution). Thus, if students already accept microevolution, then instructors may be able to more effectively persuade students to accept macroevolution and human evolution if they are able to logically articulate how microevolution leads to macroevolution and human evolution. However, this may only be effective if instructors can reduce perceived conflict with evolution and religious beliefs among Christian and Muslim students who likely see microevolution as more compatible with their religious worldviews than macroevolution or human evolution.

## Religious Cultural Competence in Evolution Education (ReCCEE) for Muslim students?

Religiosity was a greater predictor of macroevolution and human evolution acceptance than understanding of evolution among Muslim students, which was similar to patterns seen among Protestant students. Prior research with Christian students shows that using Religious Cultural Competence in Evolution Education (ReCCEE) when teaching evolution can help reduce students' perceived conflict between their religious beliefs and evolution [57–59]. Cultural competence is the ability of one culture to effectively communicate to another culture and was born from healthcare studies to take into account racial/ethnic differences between physicians and their patients [60]. In prior research, we adapted this framework to consider cultural differences between secular instructors and Christian students, but this framework could also be useful for non-Muslim instructors who are addressing Muslim student religious beliefs while teaching evolution [9]. The following is a list of practices outlined in ReCCEE that have been adapted to be potentially useful when teaching evolution to Muslim students. While we do not know of studies that have explicitly tested the efficacy of these practices for Muslim student acceptance of evolution, we propose that these are ripe areas for future research to increase Muslim student interest in evolution and their acceptance of evolution. Below we list instructional strategies from the ReCCEE framework that could be explored with Muslim students learning evolution. For a more in-depth overview of the ReCCEE framework, how it was created, and the instructional strategies included in the framework, see Barnes & Brownell, 2017 [9].

**Provide examples of Muslim scientists that accept evolution.** Prior research shows that when religious students are provided with role models who reflect their identity when learning

evolution, it can help them to accept evolution [57, 61, 62]. Instructors can provide Muslim students with examples of scientists who study evolution who are also Muslim. For instance, Fatimah Jackson is a Muslim biologist and anthropologist who won the Charles R. Darwin Lifetime Achievement Award from the American Association of Physical Anthropologists (https://www.physanth.org/news/aapa-announces-2020-darwin-lasker-and-communica-tionoutreach-awardees/charles-r-darwin-lifetime-achievement-award-2020-fatimah-jackson/) and Rana Dajani is a molecular biologist who has written about the compatibility between Islam and evolution (https://evokeproject.org/1269-2/) [63]. Highlighting these scholars and their role in evolutionary thinking may help Muslim students to see that their religious beliefs do not have to necessarily conflict with evolution.

**Teach the scientific process and evolutionary biology as agnostic rather than atheistic.** Many students come into the college biology classroom perceiving that in order to fully accept evolution, one would have to be an atheist [5, 57] and this perception is prevalent among both religious and non-religious college biology students [5]. Further, among highly religious students, this perception of evolution as "atheistic" is related to lower levels of evolution acceptance [5]. However, science, including the science of biology, can be accurately described as agnostic. Thomas Henry Huxley, also known as "Darwin's bulldog" in the nineteenth century coined the term agnostic to describe the most scientific stance on supernatural claims [64–66]:

> Agnosticism is of the essence of science . . . It simply means that [we] shall not say [we] know or believe that which [we] have no scientific grounds for professing to know or believe . . . Consequently, agnosticism puts aside not only the greater part of popular theology, but also the greater part of anti-theology . . . Agnosticism simply says that we know nothing of what may be beyond phenomena. (Huxley, 1884)

To help students overcome this misperception that evolution makes claims about the existence of a God/god(s), instructors can teach the limits of scientific knowledge as explaining the natural world and explicitly describe evolution as agnostic with respect to a God/god(s) rather than atheistic [5, 67] so that it does not have to conflict with their belief in God/god(s).

**Discuss potential compatibility between religion and evolution.** The practices outlined in the ReCCEE framework, wholistically, aim to highlight areas of potential compatibility between religion and evolution. For instance, although a literal interpretation of some religious texts is not compatible with evolution (e.g., the special creation of humans separate from other animals), often it is possible for students to reconcile their religious beliefs with evolution if they interpret creation stories in religious texts as symbolic. Overall, the evolution education literature suggests that discussing these areas of potential compatibility between religion and evolution will be effective at increasing student acceptance of evolution [58, 62]. However, students and instructors report that evolution instructors often either ignore religion when teaching evolution or they only discuss where evolution and religion are in conflict [68, 69]. When instructors ignore religion, students may assume that religion and evolution have to be in conflict [68] and when instructors highlight only the conflict between religion and evolution, students cite this as a barrier for their learning of evolution [68]. But when instructors present religion and evolution as reconcilable, student acceptance of evolution increases [70]. Thus, if instructors are interested in increasing Muslim students' acceptance of evolution, they may need to discuss the relationship between science and religion and highlight areas in which there is potential compatibility between religion and evolution [71].

**Future research.**   We give recommendations for how to improve the experiences of Muslim students using the Religious Cultural Competence in Evolution Education (ReCCEE) framework, but this framework was built from studies that are largely composed of Christian participants. Future research should explore the use of cultural competence specifically for Muslim students who have a distinct religious background and culture from Christian students. We can only know the impact of these practices on Muslim students if researchers continue to explore the unique experiences of Muslim students in evolution education.

## Limitations

We gathered data from a large number of courses and states in different geographic regions to try and create a representative sample of introductory college biology students. However, similar to most education research studies, we had to use a convenience sampling procedure and thus the results may not be generalizable to the broader population of introductory biology students. Of note, we recruited students from only 13 states, so while this was a national approach, the experiences of Muslims in the other 37 states could be different. However, given how few research papers have been published on Muslim students in the United States, these data from 13 states are a valuable starting point for understanding U.S.-based Muslim student evolution perceptions.

This study is limited by the quantitative nature of the study. Although we are able to look at averages of variables related to Muslim students' evolution education experiences, we were not able to get a more detailed understanding of (1) how these students developed their views on evolution, (2) who or what specific experiences were influential for determining these students' views on evolution, and (3) how Muslim students who have high acceptance of evolution came to their current conceptions. All of these insights would be helpful for making concrete recommendations for instructors when teaching evolution to Muslim students. Future interview studies could help illuminate specific experiences that are influential for Muslim students and help determine how to best make evolution instruction most inclusive for these students.

## Conclusion

We found that Muslim students tend to accept evolution less and are interested in evolution less than Catholic, Jewish, Hindu, Buddhist, agnostic, and atheist students. Muslim students had particularly low human evolution acceptance levels and only 36.5% of Muslim students thought life on earth shared a common ancestor. Religiosity and understanding of evolution were important predictors of evolution acceptance among Muslim students (as well as students in the broader population) and higher religiosity was a particularly strong negative predictor of human evolution acceptance and acceptance of the common ancestry of life. These findings indicate that if instructors are interested in creating more inclusive environments for Muslim students or if they are interested in increasing these students' acceptance of evolution, they may need to consider the religious beliefs and cultures of Muslim students in their classes while teaching evolution.

## Supporting information

**S1 File.**
(DOCX)

## Acknowledgments

We would like to thank the instructors of the courses included in this study who were willing to send our survey to their students and the students for completing the survey. We would also like to thank Rachel Scott for her editing of the manuscript.

## Author Contributions

**Conceptualization:** M. Elizabeth Barnes, Sara E. Brownell.

**Data curation:** M. Elizabeth Barnes, Julie A. Roberts.

**Formal analysis:** M. Elizabeth Barnes.

**Funding acquisition:** M. Elizabeth Barnes, Sara E. Brownell.

**Investigation:** M. Elizabeth Barnes, Julie A. Roberts, Samantha A. Maas, Sara E. Brownell.

**Methodology:** M. Elizabeth Barnes, Sara E. Brownell.

**Project administration:** M. Elizabeth Barnes, Sara E. Brownell.

**Resources:** Sara E. Brownell.

**Software:** M. Elizabeth Barnes.

**Supervision:** M. Elizabeth Barnes, Sara E. Brownell.

**Validation:** M. Elizabeth Barnes, Sara E. Brownell.

**Visualization:** M. Elizabeth Barnes.

**Writing – original draft:** M. Elizabeth Barnes.

**Writing – review & editing:** M. Elizabeth Barnes, Julie A. Roberts, Samantha A. Maas, Sara E. Brownell.

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
