## [Decision Letter · Decision Letter 0]

26 Apr 2021

PONE-D-21-01450

Muslim Undergraduate Biology Students’ Evolution Acceptance in the United States

PLOS ONE

Dear Dr. Barnes,

Thank you for submitting your manuscript to PLOS ONE. After careful consideration, we feel that it has merit but does not fully meet PLOS ONE’s publication criteria as it currently stands. Therefore, we invite you to submit a revised version of the manuscript that addresses the points raised during the review process.

I apologize for the delay in reviews!  It was rather difficult to find reviewers.  I suspect COVID has somewhat to do with this.  This manuscript shows great promise and I thoroughly enjoyed it.  I believe the reviews are fair and thorough and warrant your attention.  I don't think they will be too burdensome.  I recommend that you address each comment from both reviewers.  Of particular note is Reviewer 2's suggestion that you expand the manuscript to include the other non-Christian religions surveyed.  I think the data very much warrants this and makes this work even more applicable. Given the very similar sample sizes, it makes sense to combine results into one manuscript targeting at least three non-Judeo-Christian religions.  In addition, both reviewers pointed out some points of bias or speculation so make sure to address those, as well.  I look forward to reading your revision!

We look forward to receiving your revised manuscript.

Kind regards,

Jamie L. Jensen, Ph.D.

Academic Editor

PLOS ONE

Journal Requirements:

Reviewers' comments:

Reviewer's Responses to Questions

**Comments to the Author**

1. Is the manuscript technically sound, and do the data support the conclusions?

Reviewer #1: Yes

Reviewer #2: Yes

2. Has the statistical analysis been performed appropriately and rigorously? 

Reviewer #1: I Don't Know

Reviewer #2: Yes

3. Have the authors made all data underlying the findings in their manuscript fully available?

Reviewer #1: Yes

Reviewer #2: Yes

4. Is the manuscript presented in an intelligible fashion and written in standard English?

Reviewer #1: Yes

Reviewer #2: Yes

5. Review Comments to the Author

Reviewer #1: * When considering relevant prior research on other countries, it might be helpful to chase down some of the research done in McGill, Canada, often associated with Brian Alters. They examined responses to evolution and evolution in education among some Canadian Muslim populations. These might be particularly relevant because I think Canadian Muslim populations are more similar to the US Muslim population than the European and Muslim-majority country research referred to in this paper.

* There is some repetition in the text that needs to be cleaned up. I started getting annoyed at reading multiple variations on "Muslims have low acceptance of evolution" and "not much is known about Muslim students in the US."

* Lines 127-132, about de-emphasis of Muslim identity after 9/11. This is speculation; an opposite reaction—putting more emphasis on distinctly Muslim aspects of identity—is just as plausible. A more important consideration is that the social profile of Muslims in North America is known to be quite different than European immigrant populations and Muslim-majority countries. North American Muslims are comparatively wealthier, more educated, and in professional occupations compared to the country averages. This might result in bringing the level of evolution acceptance up compared to what might be expected otherwise. I didn't notice that the study was able to control for socioeconomic and parental occupational status; that might not have been feasible, but the fact should be noted.

* Someone who knows more about survey techniques and analysis should also review the technical aspects of this paper. I see nothing wrong offhand, but this type of research is far from my expertise.

* Line 227: I'm slightly confused—previously items were scored on a 1-5 Likert scale, but suddenly this becomes a 1-10 scale for Interest in Evolution. This might need some clarification.

* I'd like to see some more explicit acknowledgment of the limitations due to n=219 Muslim students as a sliver of a much larger data set. (There is some of this in the limitations section; it needs more emphasis.)

* Lines 462-468: The authors argue that exploring religion and evolution may increase acceptance of evolution. Why? Is there any solid evidence for such a proposition? This seems to be speculation.

* Line 501: I'd be more careful about reference (62). There is a small literature about alleged Muslim anticipations of evolution. I have invariably found such claims to rely on dubious interpretations of the history of science and a questionably broad understanding of "evolution" that erases substantial differences between premodern ideas about continuities in nature and the modern biological sense of evolution.

* Section starting at line 504. This notion of the "bounded nature of science" may have become dogma within science education circles, but it is intellectually quite dubious. Do the authors really want to commit to such a philosophically naive view? I wince to see that Gould's NOMA is a major reference in this section.

* I would question the competence of biology instructors to wade into Muslim religious beliefs and suggest a cheap compatibility between traditional beliefs and evolution. I doubt it would work well, especially with more theologically conservative Muslim students who are apt to notice religious ignorance on part of the instructor.

Reviewer #2: Overall the manuscript is well written and does not overstate any findings. It is a push forward in US evolution education and will be useful to those that educate Muslim students. The authors should be congratulated on focusing on more than Christian religions in the US. I did list the manuscript in need of "major revision." In reality it is in in between major and minor revision. It needs to have the scope addressed and some biased ideas removed.

Major criticisms:

My main criticism comes in the form of a question. Why only focus on Muslim students? While I agree there is a need to understand Muslim student views on evolution in the US I am not sure why there wasn't an attempt to also focus on Hindu and Buddhist students for this research. The sample sizes were similar and the findings were actually quite similar. A manuscript focusing on all three non-Christian religions is most appropriate and would make the work more interesting and impactful. Certainly by focusing only on Muslims the work is more focused but it does not seem that the paper would be hurt in any way by making it extend to other non-Christian religions as the analyses are done, the data a clearly there and all the discussion and conclusions could very easily be expanded to include Hindu and Buddhist students. Very little that is stated in the manuscript is specific to Muslim students. It would simply be appropriate to include all three in one single publication.

Line 457: I take the most issue with this sentence in the entire manuscript. It comes off as very non-educational and completely misses the mark. Why would an educator not use the common acceptance of microevolution across all religions as the place to start to build bridges and drive deeper acceptance across all aspects of evolutionary biology for all students? Especially, if the concern is that some students might de-emphasize who they are due to a sense of conflict in their beliefs with other students (their peers). My mind is completely blown that this sentence makes it into an educational piece, especially one that is focused on driving acceptance of evolution. As an educator we must always build on what is accepted and build bridges to what is not. Please fix this.

Lines 466-468: Not if the instructor hasn't identified and built upon what they do accept (see Line 457).

Minor comments:

Abstract:

Line 35 & 36: The first sentence should be qualified as being among US students, North American students, or Western students, but not as it is.

Line 52 & 53: Low or Lower? This also needs to be put in the context of the US as similar results have been shown outside the US prior to this publication.

Lines 115-118: Are these lines needed? The first sentence reviews what was directly reviewed only a few lines above and the last sentence has been repeated several times at this point in the document and is becoming highly repetitive.

Lines 128-129: This would be more interesting in the context of multiple non-Christian religions.

Lines 130-132: Why would they de-emphasize it? If Muslim students are surrounded by Protestant and LDS then why de-emphasize it? It may actually be a reason not to de-emphasize it.

Lines 134: Who are their non-Muslim peers? Certainly this varies between institutions but it goes to my previous point that if they are surrounded by peers that are also questioning and/or non-accepting of evolution, regardless of religion, then why de-emphasize their Muslim identity. Seems it would likely do the opposite depending on the composition of peers. Being non-accepting of evolution as an Muslim among a group non-accepting Christians should make Muslims more average and thus more accepted.

Lines 139 and 140: This section is the most novel portion of this research and should be the major focus.

Line 153-156: Or other non-Christian groups and a reason to treat them all in this manuscript.

Line 255: There is an assumption here that if students were born in the US they were raised here. Additional questions about what their current age and when they arrived in the US would be helpful. Another question that would seem to be very important is if their parents were immigrants as this would allow the authors to tease apart acceptance of Muslim students over generations. Not central to the current manuscripts scope but important to future research.

Line 437: in the US or elsewhere?

Lines 451-468: It seems some discussion has seeped into this section of the results.

Lines 471-482: This seems like introduction

Lines 482-483: This line seems awkward and is restated in parts in the rest of the language in the paragraph.

Lines 537-555: Could these be moved to the materials and methods? It would make for a cleaner read and a more positive ending to the manuscript.

6. PLOS authors have the option to publish the peer review history of their article (what does this mean?). If published, this will include your full peer review and any attached files.

Reviewer #1: **Yes: **Taner Edis

Reviewer #2: No

---

## [Author Response · Author response to Decision Letter 0]

11 Jun 2021

Reviewer #1: * When considering relevant prior research on other countries, it might be helpful to chase down some of the research done in McGill, Canada, often associated with Brian Alters. They examined responses to evolution and evolution in education among some Canadian Muslim populations. These might be particularly relevant because I think Canadian Muslim populations are more similar to the US Muslim population than the European and Muslim-majority country research referred to in this paper.

Thank you for this suggestion. We were able to find papers conducted on Muslim individuals by Brian Alters (one of which was previously included in the manuscript), but they were all outside of Canada. We did find one paper, not by Brian Alters, but by Anila Asghar, which included some data on Canadian Muslim High School teachers, and we have now included that in the manuscript. 

* There is some repetition in the text that needs to be cleaned up. I started getting annoyed at reading multiple variations on "Muslims have low acceptance of evolution" and "not much is known about Muslim students in the US."

Thank you for noticing this! We tried to highlight the novelty of the study to make it apparent, but we agree it became repetitive and we have removed several instances of this repetition.

* Lines 127-132, about de-emphasis of Muslim identity after 9/11. This is speculation; an opposite reaction—putting more emphasis on distinctly Muslim aspects of identity—is just as plausible. A more important consideration is that the social profile of Muslims in North America is known to be quite different than European immigrant populations and Muslim-majority countries. North American Muslims are comparatively wealthier, more educated, and in professional occupations compared to the country averages. This might result in bringing the level of evolution acceptance up compared to what might be expected otherwise. I didn't notice that the study was able to control for socioeconomic and parental occupational status; that might not have been feasible, but the fact should be noted.

Thank you for this comment and suggestion. We agree that there are more substantial factors that may contribute to higher evolution acceptance among Muslim students in the US compared to students in other countries. We have removed the reference to potential de-emphasizing of the Muslim identity and added references that discuss how Muslims in the United States tend to (1) have a higher education levels and (2) are less likely to think the Quran should be taken literally. 

* Someone who knows more about survey techniques and analysis should also review the technical aspects of this paper. I see nothing wrong offhand, but this type of research is far from my expertise.

* Line 227: I'm slightly confused—previously items were scored on a 1-5 Likert scale, but suddenly this becomes a 1-10 scale for Interest in Evolution. This might need some clarification.

Thank you for the question! The measures that are scored on a 1-5 Likert scale were not created by the authors, but previously published surveys with validity evidence. So, we did not design the response scale for these measures. However, a 5pt Likert scale is a generally accepted way to measure underlying latent constructs in survey research. There were not previous measures published for interest in evolution and we created this measure for a larger study. We used a 10 pt scale because it allowed for finer grained data, but this would not have any bearing on validity of the analyses, except to provide a finer grain description of participants’ interest.

* I'd like to see some more explicit acknowledgment of the limitations due to n=219 Muslim students as a sliver of a much larger data set. (There is some of this in the limitations section; it needs more emphasis.)

We agree that at first glance this might seem low, but the national population of Muslims in the United States is only about 1.1% of the total population. In our sample, students comprised about 2.8% of the total sample. So, in this sense, these students were overrepresented in our data set compared to the national composition. Further, statistically, 219 students are a large enough population to run all of the statistics we ran without running into any power issues. This is why we had to collect data from over 7,900 students in total so that we could get this substantial number of Muslim students for our analyses. Usually, for regressions, one needs at least 10 students per group, per variable in the analysis, which we had in our data set. So, we don’t think we are under sampled or under powered in our analyses. We added information about the percent of Muslims in the United States compared to our sample in the manuscript so that readers can compare.

* Lines 462-468: The authors argue that exploring religion and evolution may increase acceptance of evolution. Why? Is there any solid evidence for such a proposition? This seems to be speculation.

Thank you for encouraging us to be clearer on this matter. Indeed, there is published evidence that discussing religion and evolution while teaching can increase acceptance of evolution and reduce perceived conflict with evolution and religion. We have clarified this in the discussion by specifically saying prior research has shown this and citing these papers.

* Line 501: I'd be more careful about reference (62). There is a small literature about alleged Muslim anticipations of evolution. I have invariably found such claims to rely on dubious interpretations of the history of science and a questionably broad understanding of "evolution" that erases substantial differences between premodern ideas about continuities in nature and the modern biological sense of evolution.

Thank you for your expertise on this! We removed this reference.

* Section starting at line 504. This notion of the "bounded nature of science" may have become dogma within science education circles, but it is intellectually quite dubious. Do the authors really want to commit to such a philosophically naive view? I wince to see that Gould's NOMA is a major reference in this section.

We appreciate this suggestion and agree that Gould’s NOMA is often perceived as intellectually dubious, and it is outside the scope of this manuscript to clarify Gould’s view and how others have misinterpreted it. Thus, we have rewritten this section to focus purely on the limit of scientific knowledge to the natural world and encourage readers to teach science and evolution as agnostic, rather than atheistic, as originally proposed by TH Huxley. 

* I would question the competence of biology instructors to wade into Muslim religious beliefs and suggest a cheap compatibility between traditional beliefs and evolution. I doubt it would work well, especially with more theologically conservative Muslim students who are apt to notice religious ignorance on part of the instructor.

We agree that biology instructors would likely be wholly unprepared to discuss Muslim belief systems in depth, so we are not advocating for wading into any religious beliefs. We have revised this section to be clearer that we are suggesting instructors might highlight areas of potential compatibility. For instance, when religious texts are interpreted as symbolic rather than literal interpretations. This may not work for conservative Muslim students who do read their religious text as literal, but previous research suggests this is less common among American Muslims who are more likely to agree the Quran can be interpreted multiple ways. We tried to make this clearer.

Reviewer #2: Overall the manuscript is well written and does not overstate any findings. It is a push forward in US evolution education and will be useful to those that educate Muslim students. The authors should be congratulated on focusing on more than Christian religions in the US. I did list the manuscript in need of "major revision." In reality it is in in between major and minor revision. It needs to have the scope addressed and some biased ideas removed.

Major criticisms:

My main criticism comes in the form of a question. Why only focus on Muslim students? While I agree there is a need to understand Muslim student views on evolution in the US I am not sure why there wasn't an attempt to also focus on Hindu and Buddhist students for this research. The sample sizes were similar, and the findings were actually quite similar. A manuscript focusing on all three non-Christian religions is most appropriate and would make the work more interesting and impactful. Certainly, by focusing only on Muslims the work is more focused, but it does not seem that the paper would be hurt in any way by making it extend to other non-Christian religions as the analyses are done, the data a clearly there and all the discussion and conclusions could very easily be expanded to include Hindu and Buddhist students. Very little that is stated in the manuscript is specific to Muslim students. It would simply be appropriate to include all three in one single publication.

We chose to make the focus is on Muslim students because of the documented perceived conflict between religion and evolution in the Muslim community coupled with the lack of data on US based Muslim students. However, we agree that the data from other religions is valuable and should be explored. Thus, we now ran our analyses for each religious affiliation, reported these results briefly in the results section, and added a paragraph in the discussion about these results. We did want to keep the main focus on Muslim students, so we put the detailed quantitative analyses of these students in the supplement.

Line 457: I take the most issue with this sentence in the entire manuscript. It comes off as very non-educational and completely misses the mark. Why would an educator not use the common acceptance of microevolution across all religions as the place to start to build bridges and drive deeper acceptance across all aspects of evolutionary biology for all students? Especially, if the concern is that some students might de-emphasize who they are due to a sense of conflict in their beliefs with other students (their peers). My mind is completely blown that this sentence makes it into an educational piece, especially one that is focused on driving acceptance of evolution. As an educator we must always build on what is accepted and build bridges to what is not. Please fix this.

We very much appreciate this comment, which helped us reconsider how we can use acceptance of microevolution as a place for students to start with and agree on their acceptance. We have taken care to rewrite this entire section with this valuable perspective in mind. We didn’t mean to imply that acceptance of microevolution is not important in the original writing, but that since it is already highly accepted it might not be a good target to increase acceptance, since it is already high. However, this idea of using shared microevolution acceptance as a starting point we believe is valuable and have now incorporated into the discussion. We believe this has strengthened the discussion. 

Lines 466-468: Not if the instructor hasn't identified and built upon what they do accept (see Line 457).

We have removed this section of the text. 

Minor comments:

Abstract:

Line 35 & 36: The first sentence should be qualified as being among US students, North American students, or Western students, but not as it is.

We agree and have added this qualifier to the first sentence.

Line 52 & 53: Low or Lower? This also needs to be put in the context of the US as similar results have been shown outside the US prior to this publication.

We again agree and have added this qualifier to the abstract.

Lines 115-118: Are these lines needed? The first sentence reviews what was directly reviewed only a few lines above and the last sentence has been repeated several times at this point in the document and is becoming highly repetitive.

We agree! Thank you for helping us make the manuscript more concise. We have removed these lines.

Lines 128-129: This would be more interesting in the context of multiple non-Christian religions.

Lines 130-132: Why would they de-emphasize it? If Muslim students are surrounded by Protestant and LDS then why de-emphasize it? It may actually be a reason not to de-emphasize it.

Lines 134: Who are their non-Muslim peers? Certainly this varies between institutions but it goes to my previous point that if they are surrounded by peers that are also questioning and/or non-accepting of evolution, regardless of religion, then why de-emphasize their Muslim identity. Seems it would likely do the opposite depending on the composition of peers. Being non-accepting of evolution as an Muslim among a group non-accepting Christians should make Muslims more average and thus more accepted.

For previous three comments on lines 128 – 134: Following concerns from both reviewers, we have removed the references related the previous three comments from the reviewer (lines 128 – 134) and instead now discuss more relevant considerations about differences between American Muslims and Muslims in other countries that may lead to higher evolution acceptance such as a willingness to interpret the Quran in multiple ways and their higher average education levels.

Lines 139 and 140: This section is the most novel portion of this research and should be the major focus.

We agree that this section is important, and we have emphasized this in the introduction and more in the discussion as well as expanded our analyses to students from each religious affiliation and not just Muslim students.

Line 153-156: Or other non-Christian groups and a reason to treat them all in this manuscript.

We agree and have added analyses of students from other religions to the manuscript.

Line 255: There is an assumption here that if students were born in the US they were raised here. Additional questions about what their current age and when they arrived in the US would be helpful. Another question that would seem to be very important is if their parents were immigrants as this would allow the authors to tease apart acceptance of Muslim students over generations. Not central to the current manuscripts scope but important to future research.

Great point! We controlled for whether the student was born in the United States in our analyses. However, your point is well taken that future research could explore how age arriving in the United States and if their parents were immigrants impacts their levels of acceptance of evolution. 

Line 437: in the US or elsewhere?

We have clarified that we mean Muslim students in the United States.

Lines 451-468: It seems some discussion has seeped into this section of the results.

We reviewed lines 451 – 468 and they are part of the discussion section and not the results section.

Lines 471-482: This seems like introduction

Thank you for this comment. We have saved this section for the discussion because we felt like it makes more sense to introduce strategies for increasing acceptance after we have established that Muslim students have low acceptance – because if they didn’t, then the instructors would not need these strategies. 

Lines 482-483: This line seems awkward and is restated in parts in the rest of the language in the paragraph.

Thanks for catching that! We agree and have revised this section to be more concise. 

Lines 537-555: Could these be moved to the materials and methods? It would make for a cleaner read and a more positive ending to the manuscript.

We appreciate this comment but after trying to move this section to the methods, it seemed out of place and so we have decided to keep this portion of the manuscript towards the end.

6. PLOS authors have the option to publish the peer review history of their article (what does this mean?). If published, this will include your full peer review and any attached files.

Do you want your identity to be public for this peer review? For information about this choice, including consent withdrawal, please see our Privacy Policy.

Reviewer #1: Yes: Taner Edis

Reviewer #2: No

---

## [Decision Letter · Decision Letter 1]

21 Jul 2021

Muslim Undergraduate Biology Students’ Evolution Acceptance in the United States

PONE-D-21-01450R1

Dear Dr. Barnes,

We’re pleased to inform you that your manuscript has been judged scientifically suitable for publication and will be formally accepted for publication once it meets all outstanding technical requirements.

Kind regards,

Jamie L. Jensen, Ph.D.

Academic Editor

PLOS ONE

Additional Editor Comments (optional):

Reviewers' comments:

Reviewer's Responses to Questions

**Comments to the Author**

1. If the authors have adequately addressed your comments raised in a previous round of review and you feel that this manuscript is now acceptable for publication, you may indicate that here to bypass the “Comments to the Author” section, enter your conflict of interest statement in the “Confidential to Editor” section, and submit your "Accept" recommendation.

Reviewer #1: All comments have been addressed

Reviewer #2: All comments have been addressed

2. Is the manuscript technically sound, and do the data support the conclusions?

Reviewer #1: Yes

Reviewer #2: Yes

3. Has the statistical analysis been performed appropriately and rigorously? 

Reviewer #1: Yes

Reviewer #2: Yes

4. Have the authors made all data underlying the findings in their manuscript fully available?

Reviewer #1: Yes

Reviewer #2: Yes

5. Is the manuscript presented in an intelligible fashion and written in standard English?

Reviewer #1: Yes

Reviewer #2: Yes

6. Review Comments to the Author

Reviewer #1: Lines 135-137: You don’t mean “human evolution” in both instances in the same sentence, do you? Please fix this; it’s confusing.

Reviewer #2: I have no further comments. Thank you for taking the time to reshape the manuscript and address all comments.

7. PLOS authors have the option to publish the peer review history of their article (what does this mean?). If published, this will include your full peer review and any attached files.

Reviewer #1: **Yes: **Taner Edis

Reviewer #2: No

---

## [Editor Report · Acceptance letter]

30 Jul 2021

PONE-D-21-01450R1 

Muslim Undergraduate Biology Students’ Evolution Acceptance in the United States 

Dear Dr. Barnes:

I'm pleased to inform you that your manuscript has been deemed suitable for publication in PLOS ONE. Congratulations! Your manuscript is now with our production department. 

Kind regards, 

on behalf of

Dr. Jamie L. Jensen 

Academic Editor

PLOS ONE